# Neural Annealing and Visualization of Autoregressive Neural Networks in the Newman–Moore Model

Estelle M. Inack [1,2,3,*], Stewart Morawetz [4] and Roger G. Melko [1,4]

1 Perimeter Institute for Theoretical Physics, Waterloo, ON N2L 2Y5, Canada; rmelko@perimeterinstitute.ca
2 Vector Institute, MaRS Centre, Toronto, ON M5G 1M1, Canada
3 yiyaniQ, Toronto, ON M4V 0A3, Canada
4 Department of Physics and Astronomy, University of Waterloo, ON N2L 3G1, Canada; sgmorawe@uwaterloo.ca
* Correspondence: einack@perimeterinstitute.ca

**Abstract:** Artificial neural networks have been widely adopted as ansatzes to study classical and quantum systems. However, for some notably hard systems, such as those exhibiting glassiness and frustration, they have mainly achieved unsatisfactory results, despite their representational power and entanglement content, thus suggesting a potential conservation of computational complexity in the learning process. We explore this possibility by implementing the neural annealing method with autoregressive neural networks on a model that exhibits glassy and fractal dynamics: the two-dimensional Newman–Moore model on a triangular lattice. We find that the annealing dynamics is globally unstable because of highly chaotic loss landscapes. Furthermore, even when the correct ground-state energy is found, the neural network generally cannot find degenerate ground-state configurations due to mode collapse. These findings indicate that the glassy dynamics exhibited by the Newman–Moore model caused by the presence of fracton excitations in the configurational space likely manifests itself through trainability issues and mode collapse in the optimization landscape.

**Keywords:** artificial neural networks; neural network trainability; variational neural annealing; computational complexity; glassy dynamics





## 1. Introduction

In recent years, the performance of machine learning models has increased considerably in areas such as computer vision or natural language processing. Deep learning [1], in particular, has surpassed previously known algorithms, improving the performance of tasks such as object recognition and machine translation, to name a few. Undoubtedly, this improvement mainly came from artificial neural networks (ANNs), powerful models that capture intricate correlations present in data. Thanks to their representational power, they act as very efficient feature extraction machines whose output is meaningful information obtained from a nonlinear transformation of an input.

Inspired by the successes of ANNs in computer science, physicists have also started to use them to study problems in various branches of physics [2], such as optics, cosmology, quantum information, and condensed matter. In the latter, they are used to identify phases of matter [3–7], increase the performance of Monte Carlo simulations [8–14], and find precise representations of the ground state of quantum systems [15–18]. A particular aspect of their capacity to characterize quantum matter [19] is their ability to be used as parameterized functions to represent the underlying probability distribution of a physical system. However, it was observed that, for certain hard problems, such as those exhibiting frustration [20–23] (e.g., due to the infamous sign problem), it was difficult for ANNs to learn the correct distribution despite their expressiveness, entanglement content, or symmetries. This observation hints at a possible conservation of computational complexity

hardness, which manifests itself in the trainability of ANNs. This work investigates this issue by visualizing the ANNs' loss landscapes.

Loss landscape visualization is an approach that aims at representing the high dimensionality of ANNs in the more intuitive two- or three-dimensional spaces. It has been successfully used in the machine learning community to study the trainability and generalization of deep neural networks [24]. In particular, it was used to understand why skip connections in residual neural networks are generalizing better than vanilla convolutional neural networks. Nowadays, in the quantum computing community, it is more and more employed to benchmark different quantum circuit architectures over a variety of tasks, such as quantum optimization or quantum machine learning [25,26]. This paper uses it to study trainability in the neural annealing paradigm.

Neural annealing is a recent technique that aims at solving hard optimization problems through the variational simulation of the annealing procedure using ANNs [27]. It was shown to be more efficient than simulated annealing [28] and simulated quantum annealing [29] in finding the ground state of various spin glasses. In this work, we use it in both its classical and quantum formulations with autoregressive neural networks, to find the ground state of the Newman–Moore model on a triangular lattice [30–32]. We choose the Newman–Moore model as a benchmark because it is exactly solvable and displays several exotic features, such as glassiness without disorder, extensive ground-state degeneracy, and fractal symmetry; the latter proved to have application in quantum memories [33,34]. Furthermore, the fractal symmetry of the Newman–Moore generates immobile excitations under local Hamiltonian dynamics, which hamper both classical and quantum Monte Carlo simulations [30,35]. In this work, we show that the neural annealing dynamics of the Newman–Moore display instabilities due to the highly rugged nature of the loss landscape geometry, despite the use of ancestral sampling, which was shown to be superior compared to Metropolis Monte Carlo sampling. Ancestral sampling, also known as autoregressive sampling, is implemented via recurrent neural networks (RNNs) [36,37], which are known to be Turing complete [38], and universal function approximators [39]. However, we find that when the annealing dynamics results in the correct ground-state energy, the sampling is generally unable to capture the multi-modal distribution of degenerate ground states, hinting at a possible mode collapse in the training procedure.

The remainder of the article is organized as follows. In Section 2, we describe the classical and quantum Newman–Moore models. In Section 3, we describe the RNN ansatz, the neural annealing protocols, and the visualization method used in this work. Section 4 reports classical and quantum variational annealing results on the Newman–Moore model, along with the corresponding visualization of the loss landscape during annealing dynamics. Conclusions are reported in Section 5.

## 2. The Newman–Moore Model

In order to investigate trainability in the neural annealing method, we use as a test-bed the two-dimensional (2D) Newman–Moore model on a triangular lattice. Its Hamiltonian is given by:

$$H = \frac{J}{2} \sum_{i,j,k \text{ in } \nabla} \sigma_i \sigma_j \sigma_k, \tag{1}$$

where $\sigma_i = \pm 1$ are Ising spins located at the lattice sites $i, j, k$ of a downward-facing triangle (see Figure 1a). The parameter $J > 0$ fixes the strength of the interactions of each spin triplet, and is used as the energy scale of the system. In this work, we set $J = 1$. We consider as a computational basis the state $\sigma = (\sigma_1, \ldots, \sigma_N)$ of $N = L \times L$ spins, where $L$ is the length of the lattice.

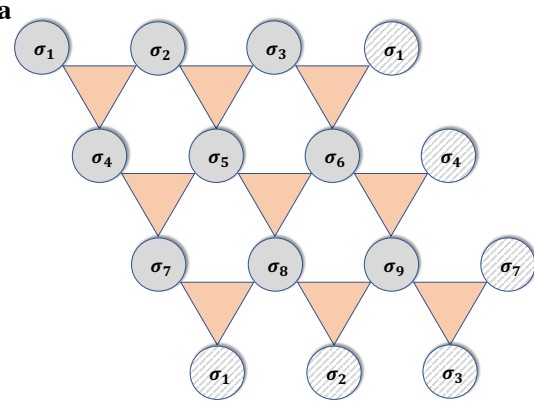

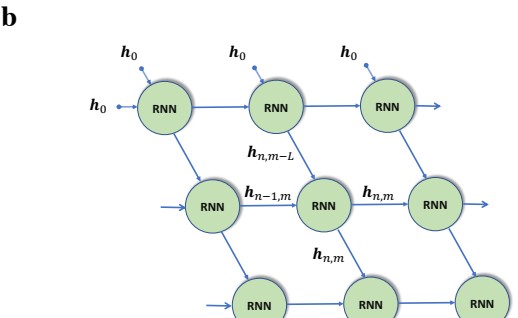

**Figure 1.** (**a**) Structure of the 2D Newman–Moore model. Spins (grey circles) interact via downward-pointing triangles. The grey-shaded spins represent periodic boundary condition interactions. (**b**) Illustration of an RNN autoregressive sampling of the 2D Newman–Moore model. An RNN cell (green circle) located at lattice site $n, m$ receives two hidden states $h_{n-1,m}$ and $h_{n,m-L}$ in a zigzag fashion (solid blue lines), as well as their corresponding spin states (not shown). $L$ is the length of the 2D lattice. The autoregressive sampling of new spins is performed sequentially along the blue lines, with the dashed blue lines indicating sampling continuation to the next row. $h_0$ represents the initial memory state of the RNN.

One peculiarity of the Newman–Moore model is that its macroscopic behavior strongly depends on the characteristics of its finite-sized lattice. If one of the sides has a dimension that can be expressed as $L = 2^k$ for some integer $k$, then the model is exactly solvable and has a single ground-state configuration, which is the trivial one with all spins pointing down. For different values of $L$, the ground state may be degenerate, with a degeneracy that is a non-analytic function of the lattice size. This is due to the Newman–Moore model exhibiting a fractal symmetry that acts on some subextensive $d$-dimensional ($d$, the fractal dimension is generally not an integer) subsystem of the whole system [33]. Furthermore, the Newman–Moore model exhibits glassy behavior at low temperatures and under single-spin-flip dynamics despite having neither randomness nor frustration. The glassiness causes a loss of ergodicity due to the presence of fracton excitations, thus hampering sampling in traditional Monte Carlo methods, even when thermal annealing is implemented [30].

Contrary to the classical Newman–Moore model, which does not have a thermodynamic phase transition at finite temperature, the quantum Newman–Moore model, in the presence of the transverse field $-\Gamma \sum_{i=1}^{N} \sigma_i^x$, exhibits a fractal quantum-phase transition at $\Gamma = 1$ [35], with first-order-like fluctuations [32]. However, fracton excitations on the

Sierpinski triangle still induce restricted mobility, hence impeding the sampling procedure of quantum Monte Carlo methods in the glassy phase [35].

In the next section, we describe our implementation of the variational classical and quantum annealing methods with recurrent neural networks.

## 3. Methods

### 3.1. The Recurrent Neural Network Ansatz

We use recurrent neural networks (RNNs) to parameterize the Boltzmann probability distribution of the Newman–Moore Hamiltonian in Equation (1). The joint probability distribution of a spin configuration $\sigma$ is modeled with $\theta$ parameters as follows:

$$p_{\theta}(\sigma) = p_{\theta}(\sigma_1)p_{\theta}(\sigma_2|\sigma_1)\cdots p_{\theta}(\sigma_N|\sigma_{<N}), \tag{2}$$

where $p_{\theta}(\sigma_i|\sigma_{<i})$ is a conditional probability given by:

$$p_{\theta}(\sigma_i|\sigma_{<i}) = \text{Softmax}(U h_{n,m} + b) \cdot \sigma_i. \tag{3}$$

The Softmax activation function guarantees that the probability distribution $p_{\theta}(\sigma)$ is normalized to unity. $\sigma_i$ is the one-hot representation of the spin $\sigma_i$, and $\cdot$ denotes the dot product operation. Note that the index $i$ covers $1, \ldots, N$. Indices $n, m$ cover $1, \ldots, L$, respectively, along the horizontal and vertical sides of the triangular lattice. $h_{n,m}$ is the RNN hidden state, which encodes information about the previous spins $\sigma_{i'<i}$. It obeys the following recurrent relation (see Figure 1b):

$$h_{n,m} = f(W^{(h)}[h_{n-1,m}; \sigma_{n-1,m}] + W^{(v)}[h_{n,m-L}; \sigma_{n,m-L}] + c), \tag{4}$$

where $f$ is a non-linear activation function of the RNN cell (green circle in Figure 1b). In this work, we use the exponential linear unit or ELU activation function. The brackets $[\ldots;\ldots]$ represent a vector concatenation operation. The parameters $U, W^{(h)}, W^{(v)}$ and $b, c$ in Equations (3) and (4) are, respectively, the weights and biases of the RNN. They are encompassed in the trainable parameters $\theta$ of the RNN ansatz.

The sampling of new spin configurations is performed autoregressively in a zigzag fashion (along the horizontal blue lines in Figure 1b) to capture the 2D structure of the lattice [18]. It is employed with the hope of sampling the degenerate ground-state configurations of the Newman–Moore model, provided that the RNN ansatz can capture its multi-modal distribution. It was shown that this approach was superior to the Markov chain Monte Carlo sampling in disordered spin glasses [27]. We used the Vanilla RNN cell in this work as we did not observe substantial improvements in using more powerful representations such as the Gated Recurrent Unit or GRU cells. We equally maintained a weight-sharing approach across all the lattice sites for the same reasons. We note, however, that more powerful representations of RNN cells, such as Tensorized RNN cells [27], could enhance the representation power of our RNN ansatz.

For the quantum case, we use the RNN ansatz in Equation (2) to model the ground-state wavefunction amplitude of the Hamiltonian as:

$$\Psi_{\theta}(\sigma) = \sqrt{p_{\theta}(\sigma)}. \tag{5}$$

The stochastic nature of the quantum Newman–Moore model allows for representing the wavefunction amplitudes with real and positive numbers instead of complex ones. For more details on RNN ansatzes, the interested reader is referred to [18,27].

### 3.2. Variational Neural Annealing

Variational neural annealing is a recently introduced method used to solve optimization problems by representing the instantaneous probability distribution of the system with neural networks during the annealing procedure [27]. Its classical formulation—

dubbed *Variational Classical Annealing* (VCA)—involves training a neural network ansatz to minimize the variational free energy at temperature $T$:

$$F = \langle H \rangle_{\boldsymbol{\theta}} - TS(p_{\boldsymbol{\theta}}). \tag{6}$$

$\langle H \rangle_{\boldsymbol{\theta}}$ stands for the ensemble averages of the Newman–Moore Hamiltonian in Equation (1). It is computed by taking Monte Carlo averages over a finite number of $N_s$ samples drawn from the RNN probability distribution $p_{\boldsymbol{\theta}}(\sigma)$. Given that $p_{\boldsymbol{\theta}}(\sigma)$ is normalized by construction, the von Neumann entropy $S(p_{\boldsymbol{\theta}}) = -\langle \log p_{\boldsymbol{\theta}}(\sigma) \rangle_{\boldsymbol{\theta}}$ is equally estimated at a moderate computational cost. The parameters $\boldsymbol{\theta}$ are trained till $F$ converges over a number of $N_{\text{warmup}}$ steps using gradient-based methods (see warmup step in Algorithm 1). Note that the variational free energy is regarded as a loss function $\mathcal{L}$ to mimic machine learning jargon. In this work, the Adam method [40] is utilized with the batch gradient descent method. The gradients of the free energy are computed efficiently using automatic differentiation, and their noise is mitigated using control-variate methods.

The temperature is slowly reduced during the annealing process according to a user-defined schedule. As is typical in simulated annealing calculations, we employ a linear annealing schedule $T(t) = T_0(1 - t)$ with $t \in [0, 1]$. Transfer learning of parameters $\boldsymbol{\theta}$ between subsequent annealing steps is implemented, as it was shown to help maintain the stability of the annealing protocol. During the annealing process, the system shifts from maximizing entropy to minimizing energy, thus finding the ground state of the Newman–Moore model provided that the rate of annealing is sufficiently slow, the ansatz is expressive enough, and the loss landscape of the free energy allows for efficient training of the RNN.

When the neural annealing algorithm is driven by quantum fluctuations instead of thermal ones, it is referred to as *Variational Quantum Annealing* or VQA. VQA is based on the Variational Monte Carlo method, where the RNN wavefunction $\Psi_{\boldsymbol{\theta}}(\sigma)$ in Equation (5) is used to approximate the ground-state wavefunction of the quantum Newman–Moore model. This is achieved by minimizing the so-called variational energy of the Hamiltonian:

$$E = \langle \Psi_{\boldsymbol{\theta}} | \hat{H} | \Psi_{\boldsymbol{\theta}} \rangle - \Gamma \langle \Psi_{\boldsymbol{\theta}} | \sum_{i=1}^{N} \sigma_i^x | \Psi_{\boldsymbol{\theta}} \rangle, \tag{7}$$

where $\hat{H}$ is the Hamiltonian operator for the Newman–Moore spin model in Equation (1). $E$ turns out to be an appropriate loss function given that it serves as an upper bound to the Hamiltonian ground energy. In VQA, the training and annealing procedures are implemented similarly to VCA, albeit at the difference of using the variational energy $E$ as a loss function with a linear annealing of the transverse field $\Gamma$. Note, however, that the computational complexity of implementing a gradient descent step is $\mathcal{O}(N^2)$ for VQA while being $\mathcal{O}(N)$ for VCA. This is due to the off-diagonal term of the transverse field. Thus, to account for larger computation costs, smaller system sizes will be used when presenting the VQA results in the next section.

A generic description of the variational neural annealing protocol (including both VCA and VQA) is displayed in Algorithm 1. More in-depth details of the neural annealing procedure can be found in Reference [27].

---

**Algorithm 1** Variational neural annealing

---

*Initialization Step:*
Randomly initialize RNN parameters $\boldsymbol{\theta}$
Set $T = T_0$ for VCA ( $\Gamma = \Gamma_0$ for VQA)
Set loss function $\mathcal{L}$ as Equation (6) for VCA ( Equation (7) for VQA)

---

*Warmup Step:*
**for** $t = 1, \ldots, N_{\text{warmup}}$ **do**
    Generate $N_s$ samples $\sigma$ from Equation (2)
    Update $\boldsymbol{\theta}$ by minimizing $\mathcal{L}$
**end for**

---

*Annealing Step:*
**do**
    $T \leftarrow T - \delta T$ ($\Gamma \leftarrow \Gamma - \delta\Gamma$ )
    **for** $t = 1, \ldots, N_{\text{train}}$ **do**
        Generate $N_s$ samples $\sigma$ from Equation (2)
        Update $\boldsymbol{\theta}$ by minimizing $\mathcal{L}$
    **end for**
**while** $T \neq 0$ ($\Gamma \neq 0$)
Generate desired samples $\sigma$ of Hamiltonian in Equation (1)

---

*3.3. Loss Landscape Visualization*

To study trainability in the neural annealing protocol, we employ a technique used to obtain a qualitative description of the loss landscape geometry in a neighborhood of the current network parameters $\boldsymbol{\theta}^*$ [24,41]. This involves plotting the loss function $\mathcal{L}$ on a randomly chosen plane given by:

$$f(\alpha, \beta) = \mathcal{L}(\boldsymbol{\theta}^* + \alpha\boldsymbol{\delta} + \beta\boldsymbol{\eta}), \tag{8}$$

where $\boldsymbol{\delta}$ and $\boldsymbol{\eta}$ are vectors of length $||\boldsymbol{\theta}^*||$ whose entries are $\mathcal{N}(0, 1)$. $\alpha$ and $\beta$ are arbitrary real numbers defining the 2D scan of the loss landscape. Although this is an arbitrary slice of the loss function high-dimensional space, it gave insight into deep neural networks' trainability and generalization properties. As shown in Algorithm 2, for each point in the vicinity of the current optimal parameters $\boldsymbol{\theta}^*$, new $N_s$ samples are generated to compute the value of the loss function at that point. This makes the landscape topography dependent on the number of samples that are used. $N_s$ is therefore fixed to the finite value used in the neural annealing simulations so that the landscape geometry reflects what the simulations experience.

Recall that the loss function in Equation (8) is, respectively, the variational free energy $F$ when VCA is implemented or the variational energy $E$ when it is VQA that is used. Thus, according to the *variational adiabatic theorem* introduced in [27], the variational annealing protocol is guaranteed to be adiabatic, provided that $\mathcal{L}$ remains convex in the vicinity of $\boldsymbol{\theta}^*$ throughout the annealing procedure.

Note that the filter-wise normalization technique of [24] is not implemented here, given that the RNN ansatz used is not scale-invariant due to the presence of ELU activation functions. A principal component analysis approach, similar to the one used in the visualization of variational quantum circuit landscapes, could also be implemented [26]. However, we found the previously described visualization method sufficient to interpret our results.

---

**Algorithm 2** Loss landscape visualization procedure

---

*Initialization:*
Set a 2D grid of $N_{\text{points}} \times N_{\text{points}}$
Set the magnitude of each point $(\alpha, \beta)$ in the 2D grid
Set the Gaussian random directions $\delta$ and $\eta$

---

**for** each point $(\alpha, \beta)$ in the 2D grid **do**
   $\overline{\theta}^* = \theta^* + \alpha\delta + \beta\eta$
   Generate $N_s$ samples $\sigma \sim p_{\overline{\theta}^*}(\sigma)$
   Compute $\mathcal{L}$ as Equation (6) for VCA (Equation (7) for VQA)
**end for**
Plot the function $f(\alpha, \beta)$ in Equation (8)

---

## 4. Results and Discussion

### 4.1. Variational Classical Annealing

In this section, we present the results of implementing the variational classical annealing method to find the ground state of the Newman–Moore model. The system is firstly equilibrated at a high temperature $T_0 = 10$ to provide enough thermal energy for exploration in the spirit of simulated annealing. This is done by minimizing the variational free energy $F$ by training the RNN ansatz over $N_{\text{warmup}} = 1000$ warmup steps. Then, annealing is performed by slowly reducing the temperature linearly over $N_{\text{annealing}} = 10{,}000$ annealing steps. These parameters were used to find the ground-state energy of the 2D Edwards–Anderson spin glass with 99.999% accuracy on a $40 \times 40$ lattice [27]; thus, we adopt them here. Details on all the hyper-parameters used in this work can be found in Table A1 in the Appendix B.

Figure 2a shows the instantaneous free energy per spin during the VCA protocol for lattices of size $L = 5, 16$ whose ground states are non-degenerate. The red curve in Figure 2a corresponds to the analytical results for the free energy density, which are exact for $L = 2^k$ [30]. As expected, the variational free energy (dark curve) values are consistently above the exact result since it acts as an upper bound to the true free energy. However, it is interesting to observe that the annealing process is not smooth, with $F$ occasionally deviating from its true value. This phenomenon is even more pronounced for the system size with $16 \times 16$ spins. This may be caused by the chaotic geometry of the instantaneous loss landscape, as shown in the next section. Even though, at the end of annealing, i.e., at $T = 0$, VCA recovers the correct ground-state energy of the Newman–Moore model (despite the simulation's instabilities in the training procedure), we have noticed that this is not always the case for different initial conditions, even, at times, for small system sizes.

Next, we perform simulations with an increased difficulty level by considering lattice sizes $L$ for which the ground state has many degenerate solutions. The results of the free energy for $L = 3, 6$ are displayed in Figure 2b. They seem to follow the same pattern as their non-degenerate counterparts, displaying a successful finding of the ground-state energy despite the somewhat chaotic annealing dynamics. Unfortunately, we have also noticed that different initializations may end up in excited states.

Furthermore, even for runs where the Newman–Moore ground-state energy is found, we have observed that the solution is almost always the trivial ground-state configuration. At the end of annealing, even after sampling a large number of configurations autoregressively, only the trivial ground-state configuration is found. Appendix A shows the only case where we noticed that the RNN was able to capture the multi-modal distribution of the ground state. In general, it seems that the model is never able to capture the multi-modal distribution of the ground-state configurations. In every instance in which VCA was tested, annealing always concluded with the unique configuration being sampled—regardless of seeding, system size, or (large) initial temperature. It is difficult to ascertain the exact cause of this mode collapse. Nonetheless, this preference for sampling a single configuration may

parallel the frozen dynamics of conventional Monte Carlo impaired by fracton excitations, potentially translating into trainability issues in VCA.

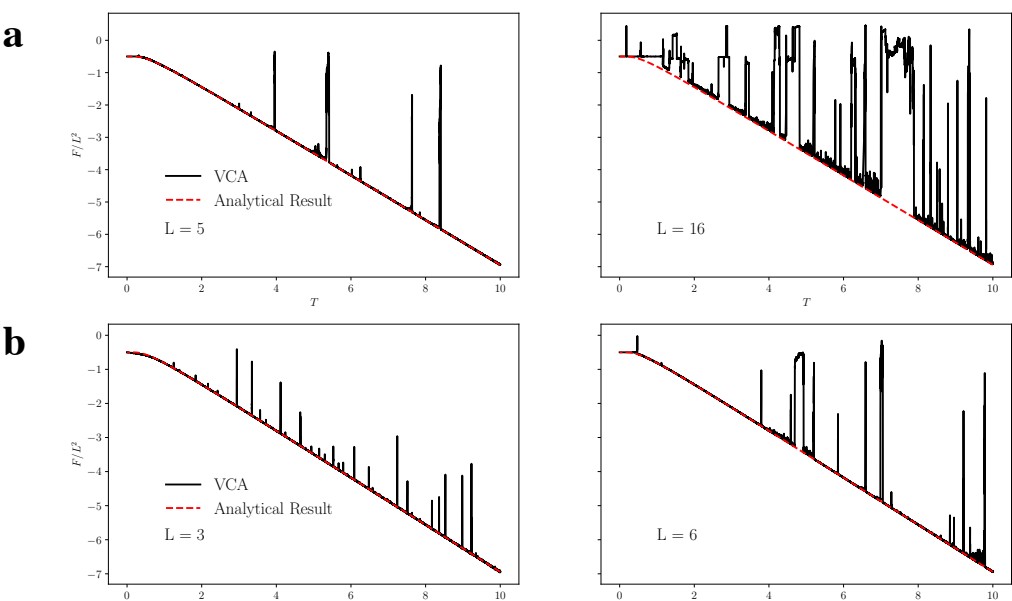

**Figure 2.** The free energy density as a function of the temperature $T$ during variational classical annealing (VCA). (**a**) System sizes $L$ depict Hamiltonians with the trivial ground-state configuration. The red dashed curve represents the exact analytical solution. (**b**) System sizes $L = 3, 6$ that represent Hamiltonians having, respectively, 4 and 16 degenerate ground-state configurations [35].

### 4.2. Variational Quantum Annealing

We now investigate the behavior of variational quantum annealing in finding the ground state of the Newman–Moore model. For comparison purposes with VCA, we perform simulations on degenerate ($L = 3$) and non-degenerate lattice ($L = 5$) sizes for which exact results are available. The exact results are obtained using the Lanczos algorithm. We equally use the same rate of annealing as in VCA. This is done by equilibrating the system at a large value of the transverse field $\Gamma_0 = 10$ before its linear decrease over $N_{\text{annealing}} = 10{,}000$ annealing steps.

Figure 3 displays the instantaneous variational energy per spin during the annealing process. It is interesting to note that, in contrast to VCA (see Figure 2), the annealing dynamics is in general much smoother. As shown in the insets, a significant jump is observed after the quantum phase transition ($\Gamma < 1$); then, the variational energy $E$ falls back to its exact value a few steps before annealing ends. However, this does not seem to affect the solution of the optimization problem as we have noticed that, for the same number of runs, VQA and VCA in general find the ground-state energy the same number of times. A possible reason for this is could be the fast convergence of directly optimizing the Newman–Moore model in Equation (1) for the system sizes used in VQA (data not shown).

Note that, despite finding the correct ground-state energy, VQA suffers from the same mode collapse issues as VCA. It often displays a strong preference for sampling only the trivial configuration once annealing is completed, regardless of system size, initial seed, or large initial transverse field (except for the example discussed in Appendix A). We equally notice that penalizing the trivial ground state does not solve the mode collapse issue, resulting in another spin configuration's mode collapse. Again, it seems that the presence of topological quasi-particle excitations in the configurational space of the quantum Newman–Moore model, which hinders practical QMC simulations, somehow translates into the inability of the variational energy loss function to capture the multi-modal distribution of the ground-state configurations.

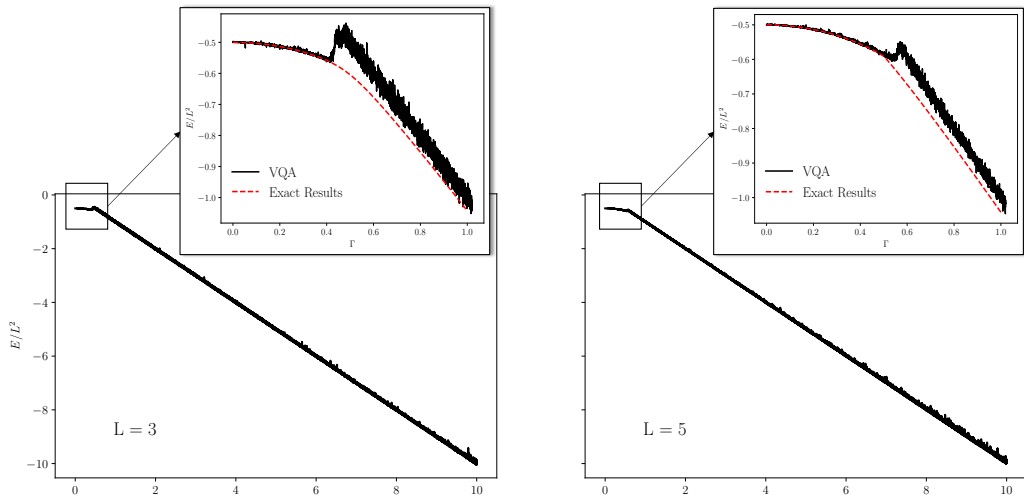

**Figure 3.** The variational energy per spin (black curves) as a function of the transverse field $\Gamma$ during variational quantum annealing (VQA). System sizes with one ($L = 5$) and multiple ($L = 3$) classical ground-state configurations are considered. The red dashed curve shows exact ground-state energies obtained from the Lanczos algorithm. The insets display VQA dynamics after the fractal quantum phase transition at $\Gamma = 1$ [35].

### 4.3. Loss Landscapes

To further understand the issues encountered during VCA and VQA simulations, we probe their loss landscapes during their neural annealing process. In Figure 4a, we show the variational free energy density landscape at the beginning, middle, and end of annealing of the simulation previously shown in Figure 2a. For each annealing snapshot, the loss landscape is plotted in the vicinity of the RNN current parameters $\theta^*$, in two random directions, as elaborated in Section 3.3. A system size of $5 \times 5$ spins is considered. Note that we have not observed significant qualitative differences between landscapes of degenerate and non-degenerate lattice sizes. The free energy is rescaled between $[-1, 1]$, as indicated by the color bar.

A couple of interesting qualitative phenomena are observable here. Before annealing at $T = 10$, there is a characteristic bright spot at the center of the panel, indicating that the current RNN parameters likely represent the global minimum of $F$. Thus, it points to a successful implementation of the warmup step. Mid-annealing seems to retain the same feature, although with a slightly reduced radius of the bright spot. This is somewhat expected given that, for $p_{\theta*}(\sigma)$ to represent the Boltzmann distribution at each temperature, the curvature around the minimum should retain its convexity. Note that some regions of low free energy separated by barriers of higher free energy are visible. This might explain the sharp deviation sometimes observed during the annealing dynamics in Figure 2a. After annealing has been completed, the landscape displays a broad plateau of constant free energy with a distinctive delimitation with a higher energy plateau and rapidly oscillating barriers. This corresponds to parameter regimes in which the same configuration is sampled exclusively and is likely the signature of the strong preference that the network has for sampling a single configuration at $T = 0$.

In Figure 4b, loss landscapes corresponding to VQA simulations in Figure 3 (for $L = 5$) are displayed. The first two panels' qualitative behaviors are similar to the ones of VCA. At mid-annealing, though, we observe a large region of constant low energy close to $\theta^*$, which may evolve to local minima as the transverse field is reduced. This may explain why the variational energy sometimes gets excited out of its exact course. The snapshot at $\Gamma = 0$ shows that around $\theta^*$, the variational energy has a local minimum separated by high-energetic barriers. This feature should be expected for a successful VQA run. We equally note the presence of another region of low energy, which might point to a regime of parameters for which low-lying excited states are sampled. However, we note

that, in general, we did not observe such a trend in other VQA runs; thus, we can only remain speculative in our observation. Furthermore, it is important to mention that, from the plots in Figure 4, it is difficult to ascertain why the VCA simulations seem more chaotic than VQA simulations. Analyzing the low-dimensional optimization trajectories during annealing might shed more light on this issue and is worth further investigation.

**a**

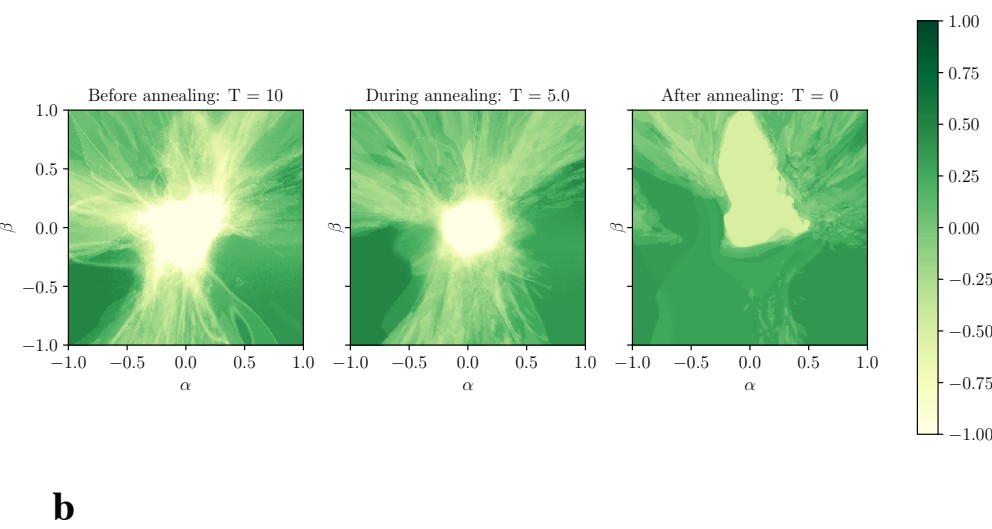

**b**

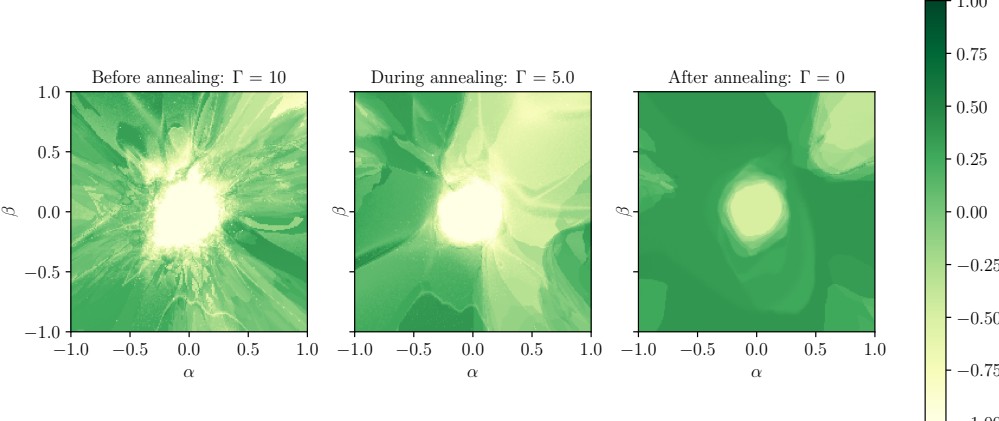

**Figure 4.** Loss landscapes of the neural network ansatzes around the optimal parameters at different snapshots of the variational annealing protocols. $\alpha$ and $\beta$ are parameters rescaling random directions around the current optimal parameters. The system has $5 \times 5$ spins. The color bar represents the loss landscape, which corresponds to (**a**) the variational free energy density in VCA simulations shown in Figure 2, and (**b**) the variational energy per spin for VQA simulations in Figure 3. Both values are rescaled for comparison purposes.

Next, we look at the loss landscape geometry for larger lattices for which trainability issues were more pronounced. Figure 5 shows 3D snapshots of the loss landscape during the VCA simulations of Figure 2a. In the first panel, before the warmup step, we observe that the free energy landscape has a local minimum around the parameters $\theta^*$. This is likely a random event, given that the RNN parameters were randomly initialized. After the warmup, the landscape maintains its shape, except that the variational free energy minimum value has now converged close to the exact one at $T_0 = 10$. As the temperature is reduced, the landscape shape becomes more rugged, with the appearance of sizeable high-energy plateaus and rapidly changing barriers, eventually leading to the disappear-

ance of the local minimum into an utterly chaotic landscape at the end of annealing. Thus, from this standpoint, it is evident that trainability issues are at work here, hindering a successful application of variational neural annealing.

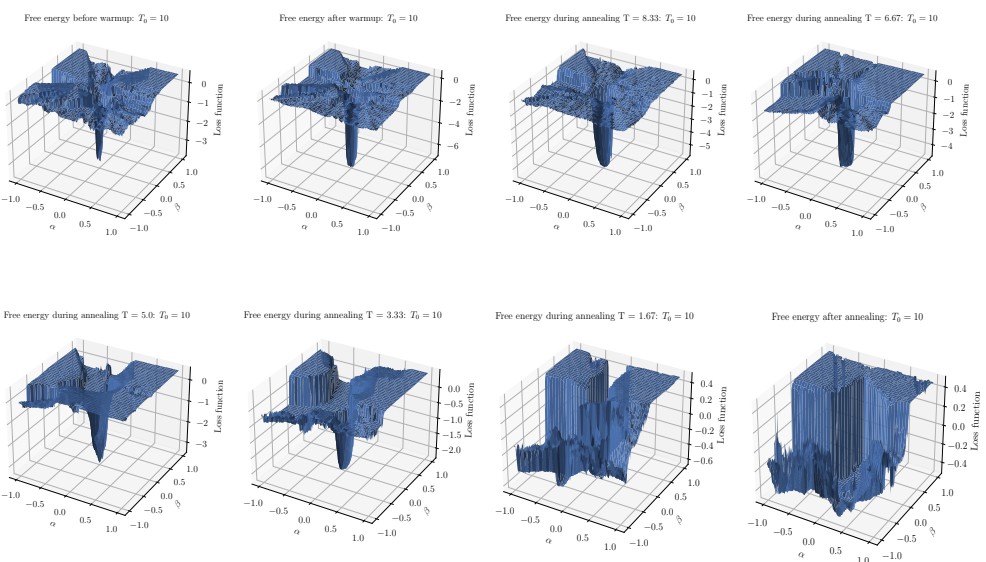

**Figure 5.** Three-dimensional snapshots of the variational free energy landscape during the VCA protocol on a $16 \times 16$ lattice. The first two snapshots display the loss landscape before and after the warmup at $T_0 = 10$. The subsequent ones depict the shape of the landscape as the temperature is annealed. The annealing dynamics corresponds to the one depicted in Figure 2a. The last snapshot is the free energy landscape at zero temperature.

## 5. Conclusions

We have implemented the variational neural annealing method using RNN ansatzes to find the ground state of the 2D Newman–Moore model on a triangular lattice. We have observed that, even when the ground-state energy was found, the neural annealing dynamics often displayed strong deviations from the instantaneous free energy for VCA, and ground-state energy for VQA, the effect being more pronounced in the former case. Furthermore, we noticed that even when VCA and VQA succeeded in finding the exact ground-state energy at the end of annealing, they consistently failed to identify the other ground-state configurations of degenerate lattices, only finding the trivial one (except for very small system sizes). These results indicate that the glassy dynamics exhibited by the Newman–Moore model due to the presence of fracton excitations likely manifests as training issues and mode collapse in neural annealing protocols.

To shed more light on our findings, we analyzed the loss landscape topologies of the VCA and VQA cost functions using a visualization technique [24] borrowed from the machine learning community. We noticed that the instabilities during the annealing protocol were caused by the chaotic geometry of the loss function, thus impeding the effective training of the RNN parameters. This result points to a potential link between glassiness in the configurational landscape of the Hamiltonian with trainability issues in the parameters space of the loss landscape. A more in-depth investigation of this phenomenon is needed.

Potential directions could be to study the optimization paths during annealing or the effect of different optimizers (e.g., stochastic reconfiguration [42]), especially those incorporating knowledge of the loss function curvature (e.g., the Hessian). Using more representative neural network architectures is also an option, even though we argue that the expressivity of the RNN ansatz used in this work is enough to represent probability amplitudes (being Turing complete) and that the simulation issues observed mainly come

from the complex loss landscapes. However, as skip connections were shown to provide a smoother landscape in convolutional neural networks [24], this is a possibility that cannot be completely ruled out. Encoding the symmetries of the Newman–Moore model in the RNN ansatz could also help to improve the simulations.

With artificial neural networks becoming standard tools to probe condensed matter systems, it is important to understand their features that contribute to efficient simulations. Symmetries, entanglement, and expressivity [43,44] are some features that have already been shown to be important. In this work, we showed that neural network learnability is also essential. Understanding when, how, and where conservation of computational complexity occurs is primordial in employing neural networks to tackle complex systems. It will provide a solid framework for which they might (or not) outperform traditional methods, which fail on significant condensed matter problems such as non-stochastic systems [22,23].

**Author Contributions:** Conceptualization, E.M.I., S.M. and R.G.M.; Data curation, E.M.I. and S.M.; Formal analysis, E.M.I. and S.M.; Funding acquisition, E.M.I. and R.G.M.; Software, E.M.I. and S.M.; Writing—original draft, E.M.I.; Writing—review and editing, E.M.I., S.M. and R.G.M. All authors have read and agreed to the published version of the manuscript.

**Funding:** Resources used in preparing this research were provided, in part, by the Province of Ontario, the Government of Canada through CIFAR, and companies sponsoring the Vector Institute www.vectorinstitute.ai/#partners. This work was made possible by the facilities of the Shared Hierarchical Academic Research Computing Network (SHARCNET) and Compute Canada. This work was supported by NSERC, the Canada Research Chair program, and the Perimeter Institute for Theoretical Physics. Research at the Perimeter Institute is supported in part by the Government of Canada through the Department of Innovation, Science and Economic Development Canada and by the Province of Ontario through the Ministry of Economic Development, Job Creation and Trade.

**Institutional Review Board Statement:** Not applicable.

**Informed Consent Statement:** Not applicable.

**Data Availability Statement:** The data are contained within the article.

**Acknowledgments:** We thank Jeremy Côté, Mohamed Hibat-Allah, Juan Carrasquilla, Sebastian Wetzel, Zheng Zhou and Andrey Gromov for the fruitful discussions. We thank Sebastiano Pilati for his valuable comments on the manuscript.

**Conflicts of Interest:** The authors declare no conflict of interest.

## Appendix A

This section provides additional neural annealing results on a degenerate system size of the Newman–Moore model. In Figure A1, we show results of VCA on a lattice of size $L = 3$ (smallest size to have degenerate ground states) corresponding to the end of annealing in Figure 2b. At the end of annealing, we observed that the 100 number of training samples are all in one of the four ground-state configurations. For statistics purposes, in Figure A1a, we show principal component analysis results on 13,222 new spin configurations sampled autoregressively from the RNN ansatz at the end of annealing. The color bar represents the Hamming distance between a new spin configuration $\sigma$ and the configuration $\sigma^*$ where all spins point down. It is given by:

$$d(\sigma, \sigma^*) = \|\sigma - \sigma^*\|_1, \tag{A1}$$

where $\|\ldots\|_1$ stands for the $L_1$ norm. We indeed observe that the RNN samples the four distinct ground-state configurations. The blue dot with Hamming distance zero represents the trivial ground state. The three other red dots represent the three other ground states, each one separated from the trivial one by six spin flips. Thus, we show that, for this case, the RNN model is able to capture the multi-modal distribution of the ground state. Note

that in Figure A1b, the distribution of the ground-state configurations is almost uniform, with a slight advantage for the first ground state (the trivial one).

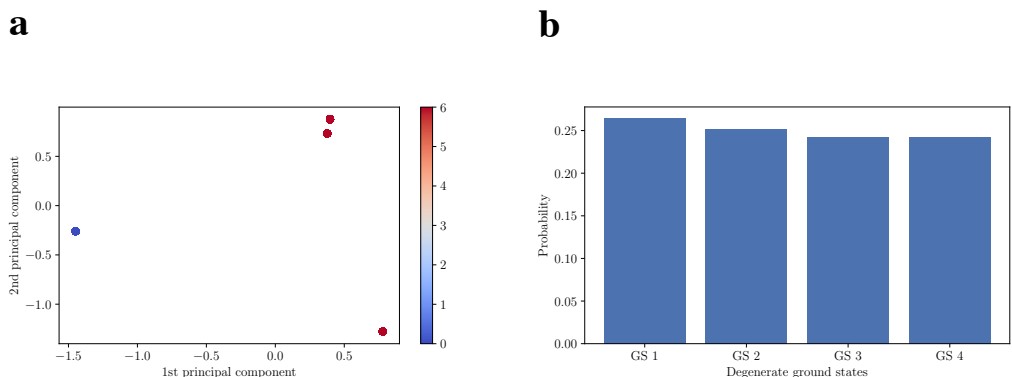

**Figure A1.** (**a**) Principal component analysis of 13,222 degenerate ground-state configurations obtained at the end of annealing of Figure 2b for $L = 3$. The color bar represents the Hamming distance between the solutions obtained from the RNN ansatz and the trivial ground-state configuration. (**b**) Probability distribution of the ground-state configurations.

VQA simulations in Figure 3 also capture the multi-modal ground-state distribution (data not shown). However, we have noticed that for both VCA and VQA, not all runs of neural annealing were able to capture all the ground states, with some finding only the trivial one. We have also observed that penalizing the loss function in VCA and VQA with the trivial ground-state magnetization often resulted in mode collapse into another ground-state configuration. It is, however, possible that a more elaborate regularization function recovers the other configurations. Furthermore, for the larger degenerate system sizes, such as $L = 6$ shown in Figure 2b, for successful runs, the simulations always resulted in the trivial ground-state configuration, even for runs with a more considerable annealing time, number of training samples, or number of hidden state variables.

## Appendix B

**Table A1.** Hyper-parameters used to perform neural annealing and loss landscape visualization.

| Hyper-Parameters | Values |
|---|---|
| Initial temperature | $T_0 = 10$ |
| Initial transverse field | $\Gamma_0 = 10$ |
| Number of warmup steps | $N_{\mathrm{warmup}} = 1000$ |
| Number of training steps | $N_{\mathrm{train}} = 5$ |
| Number of annealing steps | $N_{\mathrm{annealing}} = 10{,}000$ |
| Number of samples | $N_s = 100$ |
| Batch size | $N_b = 10$ |
| Hidden state dimension | $d_h = 40$ |
| Learning rate | $\eta = 10^{-4}$ |
| Number of grid points | $N_{\mathrm{points}} = 200$ |

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
