# Peer review of "Neural Annealing and Visualization of Autoregressive Neural Networks in the Newman–Moore Model"

_condensedmatter, doi:10.3390/condmat7020038_

Round 1

Reviewer 1 Report

The article describes possibility of using an two-dimensional Newman-Moore model on a triangular lattice for conservation of computational complexity in the learning process of neural network. Theoretical description of the problem is reasonable and the description of the algorithms is clear. The level of work is partially reduced by the fact, that used algorithm (Newman-Moore) is well known and not new and use of algorithm is mainly theoretical. The practical part is written too simply and briefly - perhaps there could be more experiments on different types of neural networks.

Notwithstanding the above, experiments, that are summarized in article, are clear and the conclusion and results are correct.

In summary, the article is well written, introduction and methods are adequately described. I recommend checking the formatting of the document (bad Figure A1 placement).

Reviewer 2 Report

The present work deal with a particular triangular interacting spin system, and search for its ground state solution through a (NN) neural network model.

After a correctly explained framework in the introduction, the section 2 describes the Hamiltonian and in the next the algorithm for the recurrent NN used is shown. In section Results... the authors show that the NN models performs reasonably only for the 0th state, although with a significant noise, and doesn't work for degenerate states at all. 

It is displayed graphicaly in Figs. 2,3,4. The conclusion are drawn in Fig.5 and some details of the method are presented in Appendix A,B. I think the calculations are well done and the problem is interesting enough to be published. I just wonder about what could improve their results: for instance, which are the hyperparameters of the NN used which gives the best solution? also, how the deviation of the true solution depends on the size of the system, either for the VQA or VCA cost functions? I suppose it is not to much extra work and will improve largely the interest in the work.
